# Low immunogenicity of LNP allows repeated administrations of CRISPR-Cas9 mRNA into skeletal muscle in mice

Eriya Kenjo [1,2], Hiroyuki Hozumi[1,2], Yukimasa Makita [1,2], Kumiko A. Iwabuchi[2,3], Naoko Fujimoto[2,3], Satoru Matsumoto[2,4], Maya Kimura[5], Yuichiro Amano[5], Masataka Ifuku[2,3], Youichi Naoe[2,3], Naoto Inukai [1,2] & Akitsu Hotta [2,3✉]

Genome editing therapy for Duchenne muscular dystrophy (DMD) holds great promise, however, one major obstacle is delivery of the CRISPR-Cas9/sgRNA system to skeletal muscle tissues. In general, AAV vectors are used for in vivo delivery, but AAV injections cannot be repeated because of neutralization antibodies. Here we report a chemically defined lipid nanoparticle (LNP) system which is able to deliver Cas9 mRNA and sgRNA into skeletal muscle by repeated intramuscular injections. Although the expressions of Cas9 protein and sgRNA were transient, our LNP system could induce stable genomic exon skipping and restore dystrophin protein in a DMD mouse model that harbors a humanized exon sequence. Furthermore, administration of our LNP via limb perfusion method enables to target multiple muscle groups. The repeated administration and low immunogenicity of our LNP system are promising features for a delivery vehicle of CRISPR-Cas9 to treat skeletal muscle disorders.

[1] T-CiRA Discovery, Takeda Pharmaceutical Company Limited, 26-1, Muraoka-Higashi 2-chome, Fujisawa, Kanagawa 251-8555, Japan. [2] Takeda-CiRA Joint Program, Fujisawa, Kanagawa, Japan. [3] Center for iPS Cell Research and Application (CiRA), Kyoto University, 53 Kawahara-cho, Shogoin, Sakyo-ku, Kyoto 606-8507, Japan. [4] Drug Product Development, Pharmaceutical Sciences, Takeda Pharmaceutical Company Limited, 26-1, Muraoka-Higashi 2-chome, Fujisawa, Kanagawa 251-8555, Japan. [5] Drug Safety Research and Evaluation, Takeda Pharmaceutical Company Limited, 26-1, Muraoka-Higashi 2-chome, Fujisawa, Kanagawa 251-8555, Japan. ✉email: akitsu.hotta@cira.kyoto-u.ac.jp

Duchenne muscular dystrophy (DMD) is a severe muscle degeneration disease caused by a loss-of-function mutation in the dystrophin gene[1]. To maintain skeletal muscle integrity, dystrophin protein plays multiple important roles in skeletal muscle. Particularly, its N-terminal domain binds to intracellular actin fibers, and its C-terminal domain forms a complex with the sarcolemma membrane. Antisense oligonucleotide (ASO) drugs, such as Eteplirsen and Viltolarsen, have been approved to induce exon skipping to restore the reading frame of the dystrophin protein in DMD patients[2]. However, since ASOs target pre-mRNA to mask the target exon from splicing, their effects are transient.

The CRISPR-Cas9 system as a genome editing tool allows us to target a desired site on a genomic sequence and to induce small deletions or insertions (indels). As such, CRISPR-Cas9 can be used to induce exon skipping at the genomic DNA level with long-lasting effects[2]. To induce genomic exon skipping in DMD patients' muscle tissue, a proper delivery vehicle is needed to introduce CRISPR-Cas9 and sgRNA targeting the human dystrophin sequence.

To this end, many groups have utilized AAV (adeno-associated virus) vectors to deliver and express Cas9 and sgRNA[3–14]. However, the long expression of AAV increases the risk of off-target mutagenesis. Indeed, recent reports have highlighted the existence of random integrations of a fragment of AAV vector into the chromosomes of a DMD mouse model[12,15]. Furthermore, the capsid protein of AAV or bacteria-derived Cas9 protein can be easily trapped by the immune surveillance system, and a second injection of AAV vector is known to be neutralized by acquired immunity[16,17]. Since it is difficult to treat large tissues such as skeletal muscle by one shot, multiple injections are assumed necessary for treatment.

Among the several non-viral delivery systems, lipid nanoparticles (LNPs) have been investigated as chemically defined RNA-delivery vehicles[18]. LNPs normally consist of 4 lipids and RNA and have a diameter of less than 100 nm. The key component for delivery is an ionizable lipid, which is an amphiphilic molecule composed of a cationic head group and two hydrophobic tails. The cationic head group, such as a tertiary amine, has no charge at neutral pH but becomes cationic at lower pH[19]. An uncharged LNP normally can bind to the cell surface via hydrophobic interactions or receptor-mediated endocytosis[20]. Once the LNP is internalized into the cell, the acidic condition in the endosome induces cationization of the ionizable lipid to break the endosome membrane and release RNA molecules into the cytoplasm[19,21]. The first siRNA drug, Onpattro™, utilizes LNP technology to deliver siRNA into hepatocytes to treat hereditary transthyretin-mediated (hATTR) amyloidosis[22]. Recently, LNPs were also developed as a vaccine platform by intramuscular injection, and two LNP-based vaccines against SARS-CoV-2 spike protein (mRNA-1273[23] and BNT162b1[24]) have been authorized for emergency use in the United States and other countries.

In the context of genome editing, recent reports demonstrated the feasibility of delivering Cas9 mRNA and sgRNA into mouse liver by LNP[25–29]. Other reports demonstrated the delivery of a Cas9 protein and sgRNA complex by LNP to target the liver, lung, muscle, and brain in mouse by adjusting the size and lipid formulation[30–32]. However, pre-existing immunity against bacterial Cas9 protein is a concern[17,33]. As for LNP technology that deliver RNA-based CRISPR-Cas9 system into broad skeletal muscle, yet to be optimized, especially in the context of multiple injections, immunological responses, in vivo safety, and injection routes.

Here, we developed an pH-dependent ionizable lipid with three hydrophobic tails and formulated it into an LNP-delivery system to preferentially target skeletal muscle. To assess CRISPR-sgRNAs that target human dystrophin gene in mouse, we generated a humanized DMD mouse model by replacing mouse exon 45 with human exon 45 and deleting mouse exon 44. The generated hEx45KI-mdx44 mice exhibited a loss of dystrophin protein, calf hypertrophy, increased plasma creatine kinase (CK) level, and regenerated myofibers with central nuclei, similar to the widely used mdx mouse model. Although ASO-mediated dystrophin recovery was transient, a single injection of LNP-CRISPR-induced durable restoration of dystrophin protein by genomic exon skipping for at least one year. Furthermore, we demonstrate that, unlike immunogenic AAV vectors, our LNP can be repeatedly administrated intramuscularly with accumulative dystrophin protein recovery. To target broader muscle tissues, we propose the injection of LNP-CRISPR via intravenous limb perfusion to induce genomic exon skipping and restoration of dystrophin protein in various muscle groups. Our LNP-CRISPR delivery methodology could open the possibility to treat local muscles of muscular dystrophy patients by repeated administrations over time.

## Results

**Optimization of LNP to target skeletal muscle by delivering Luc mRNA.** We first screened for LNP formulations that could efficiently deliver mRNA into skeletal muscle. Luciferase (Luc) mRNA was encapsulated into various formulations of LNPs (Fig. 1a), which were evaluated for their size distribution and mRNA encapsulation efficiency (Supplementary Fig. 1a, b). Based on the transfection efficacy in mice (Supplementary Fig. 1c), we selected a novel ionizable lipid, TCL053 (Fig. 1b), and an LNP formulation, LNP-I, for further assessment. To evaluate the gene expression kinetics of LNP-Luc mRNA in mouse skeletal muscle, we injected LNP-Luc mRNA into the gastrocnemius muscle (GC) of C57BL/6J mice and compared the effects to an AAV vector packaged with Luc cDNA (AAV2-Luc). The Luc expression in mice treated with high doses of AAV2-Luc was detected at 7 days post injection and lasted for nearly 100 days (Fig. 1c, d). By contrast, robust Luc expression was detected 4 h post injection by LNP-Luc intramuscular injection, but the bioluminescence rapidly decreased over time and was undetectable at 7 days, as anticipated[34] (Fig. 1c, e).

**Delivery of Cas9 mRNA and sgRNA into mouse muscle by LNP.** We then examined whether LNP can be used for CRISPR-Cas9/sgRNA delivery. Because of the size difference of Cas9 mRNA (4.2 kb) and sgRNA (100 base), we encapsulated mRNA and sgRNA separately into LNPs. When we challenged the LNPs with RNase A, the encapsulated mRNA and chemically modified sgRNA were stable for at least 24 h (Supplementary Fig. 2). To evaluate Cas9 mRNA delivery with LNP, LNP-Cas9 mRNA was injected into the GC of C57BL/6J mice. The expression kinetics of LNP-Cas9 mRNA in skeletal muscle was comparable to LNP-Luc mRNA, and the expression of Cas9 protein rapidly decreased over time (Supplementary Fig. 3a, b). To investigate the genome editing efficacy, we intramuscularly injected LNP-Cas9 mRNA and LNP-sgRNA targeting the mouse Rosa26 locus into the GC of C57BL/6J mice. Four days after the injection, we assessed the indel rate at the Rosa26 locus in GC genomic DNA and found that the co-injection of LNP-Cas9 mRNA and LNP-sgRNA induced genome editing in a dose-dependent manner (Supplementary Fig. 3c). These data demonstrate that LNP can deliver and induce CRISPR-Cas9/sgRNA-mediated genome editing in skeletal muscle tissue, albeit with the transient expression of Cas9 protein.

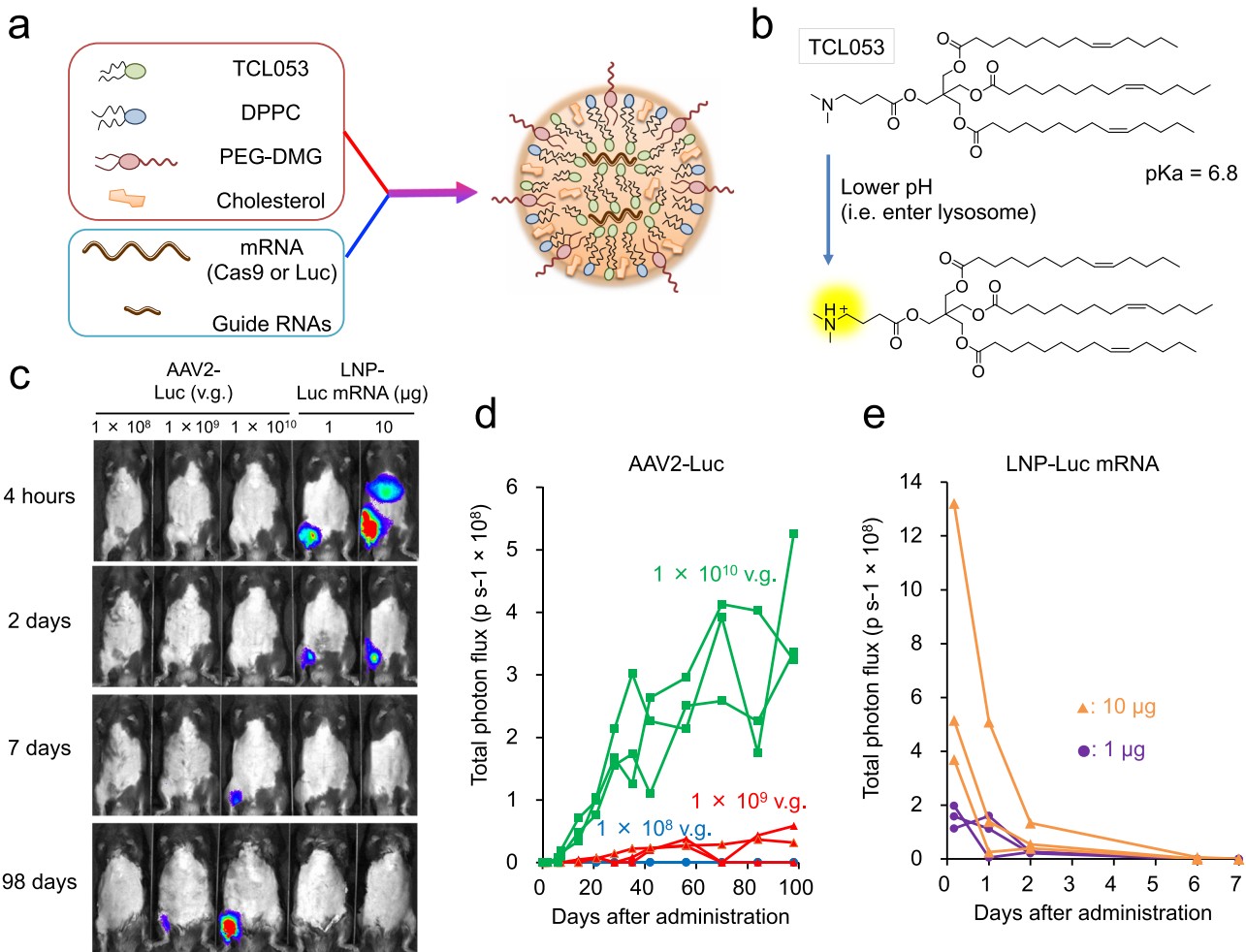

**Fig. 1 LNP-mediated Luc-mRNA or CRISPR-Cas9 mRNA/sgRNA delivery into muscle tissue. a** Schematic illustration of LNP-CRISPR. Either Luc mRNA or Cas9 mRNA/sgRNA is encapsulated into LNP that consists of TCL053, DPPC (Dipalmitoylphosphatidylcholine), PEG-DMG (Polyethylene glycol-dimyristoyl glycerol), and cholesterol. **b** Chemical structure of the newly synthesized ionizable lipid, TCL053. **c** Representative bioluminescence images of C57BL/6J mice after the intramuscular injection of AAV2-Luc ($1 \times 10^8$, $1 \times 10^9$, or $1 \times 10^{10}$ v.g., vector genomes) or LNP-Luc mRNA (1 or 10 μg mRNA). **d**, **e** Quantification of the bioluminescence signal in skeletal muscle of C57BL/6J mice treated with AAV2-Luc (**d**) or LNP-Luc mRNA (**e**). The same mice ($n = 3$ mice per group) were examined repeatedly over time. Total flux data (p s$^{-1}$, photons per second) are plotted as a single line per mouse.

**LNP with dual sgRNAs induce exon 45 skipping in DMD patient myoblasts**. We recently identified a pair of sgRNAs, hEx45 sgRNA #1 to target the splicing acceptor site and hEx45 sgRNA #23 to target the donor site of exon 45 in human *DMD* gene[35] (Fig. 2a). The dystrophin reading frame can be restored by CRISPR-Cas9-mediated exon 45 skipping in DMD patients lacking exon 44 (Fig. 2b). To verify the utility of LNP for DMD therapy, we formulated hEx45 sgRNA #1 and hEx45 sgRNA #23 into separate LNPs and evaluated the dystrophin restoration in myoblasts derived from DMD patient-induced pluripotent stem (iPS) cells. These iPS cells lacked exon 44 and were differentiated into myoblasts by MYOD1 over-expression, as previously reported[35]. LNP-Cas9 mRNA was mixed with LNP-hEx45 sgRNA #1, LNP-hEx45 sgRNA #23 or both and added to the patient-derived myoblast culture. Seventy-two hours later, dystrophin restoration was assessed by RT-PCR and Western blotting (Fig. 2c). We found that the exon skipping efficiency and dystrophin expression were higher when myoblasts were treated with both sgRNA #1 and #23 than either sgRNA alone (Fig. 2d, e).

**Potential off-target DNA cleavage sites by cell-free CIRCLE-seq analysis**. To evaluate the potential off-target cleavage sites of the

sgRNAs, we conducted a cell-free genome-wide off-target analysis, CIRCLE-seq[36], using genomic DNA from human iPS cells. We performed two independent CIRCLE-seq experiments for each sgRNA and considered common cleavage sites from the two experiments as candidate CIRCLE-seq off-target sites (Supplementary Fig. 4a). As a result, human dystrophin on-target sites for sgRNA #1 and #23 were identified as the highest peak, and several other potential off-target peaks were also detected (Supplementary Fig. 4b). In the original CIRCLE-seq paper, the authors reported that only a subset of CIRCLE-seq sites can be detected for indels in vitro or in cells, owing to the high sensitivity of the deep sequencing-based assay. Hence, we picked 6 candidate off-target sites for sgRNA #1 and 10 sites for sgRNA #23 and validated the in vitro DNA cleavage activity using PCR fragments as the substrate (Supplementary Fig. 4c). Under the condition in which 100% cleavage activity was observed for on-target sites, partial cleavage activities were detected only in 2 sites for each sgRNA (Supplementary Fig. 4c, d).

Next, to validate the potential cleavage and mutagenesis activities at the detected off-target sites in cells, we utilized Dox- and Dex-inducible, Cas9-expressing HEK293T cells[37] to mimic the high-level and long-term expression of Cas9/sgRNA. After 8 weeks of continuous Cas9/sgRNA expression,

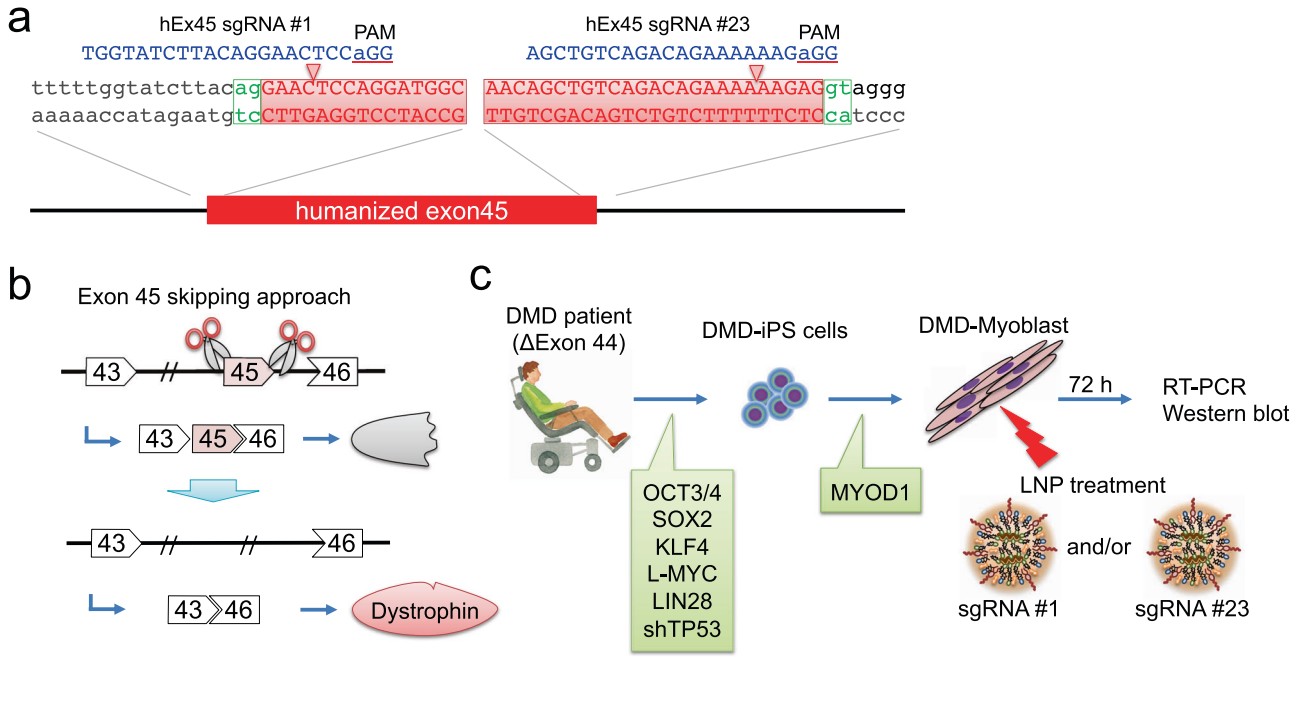

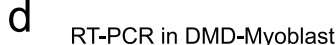

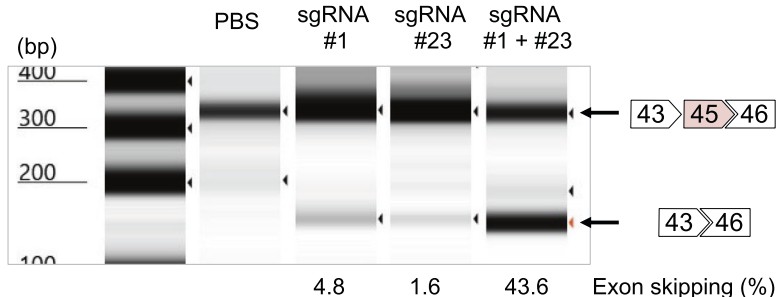

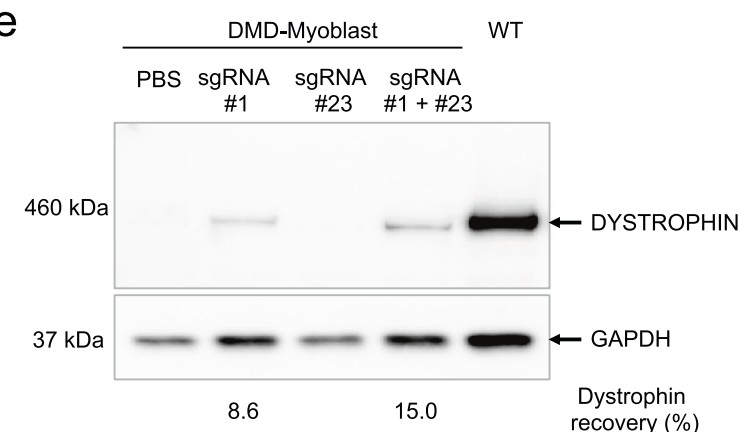

**Fig. 2 Exon 45 skipping of human dystrophin using dual sgRNAs in DMD-iPS cells. a** Two sgRNAs were designed to induce human exon 45 skipping. hEx45 sgRNA #1 and hEx45 sgRNA #23 target the splice acceptor and donor site, respectively. The human *DMD* exon 45 sequence is highlighted in red. **b** Therapeutic strategy of genomic exon skipping by the dual sgRNA approach. Exon 45 skipping results in restoration of the dystrophin protein by adjusting the protein reading frame. **c** DMD-iPS cells were derived from a DMD patient lacking exon 44. Myogenic differentiation of the DMD-iPS cells was induced by the overexpression of MYOD1 (DMD-Myoblast). LNP-CRISPR, which is a mixture of LNP-Cas9 mRNA (1 μg mRNA) and LNP-sgRNA (total 1 μg of sgRNA: either 1 μg of #1, 1 μg of #23, or 0.5 μg each of #1 and #23), was administrated for 72 h. **d** Exon skipping efficiency was measured by RT-PCR and TapeStation. *n* = 1 experiment is shown, but similar exon skipping were observed in more than 3 experiments. **e** Dystrophin restoration was measured by Western blot. *n* = 1 experiment is shown, but similar Dystrophin recovery were obtained in more than 3 experiments.

we extracted genomic DNA and assessed the presence of indels by the T7E1 assay (Supplementary Fig. 5a). Interestingly, apart from the *DMD* gene on-target site, no obvious off-target mutagenesis was detected in the candidate sites identified by the CIRCLE-seq analysis (Supplementary Fig. 5b), suggesting that the risks to introduce a mutation on the detected off-target sites are much lower compared with the on-target dystrophin cleavage activity, especially in cells.

**Replacement of mouse exon 45 with human exon 45 to generate a mouse model of DMD.** To evaluate our optimized sgRNAs targeting human exon 45 in a mouse model, we generated a novel mouse model of DMD, in which we first replaced mouse exon 45 and surrounding introns with the corresponding human sequence by knock-in (hEx45KI) and subsequently deleted mouse exon 44 by CRISPR-Cas9 (hEx45KI-mdx44) (Fig. 3a). Dystrophin transcripts were assessed in skeletal muscle, heart, and liver by RT-PCR (Fig. 3b), confirming the lack of exon 44 (148 bp) in skeletal muscle and heart. The RT-PCR bands from hEx45KI-mdx44 mice were further analyzed by Sanger sequencing, confirming the lack of mouse exon 44 and proper splicing of humanized exon 45 (Fig. 3c). By a Western blot of skeletal muscle and heart samples, a comparable molecular weight of dystrophin protein from wild-type mice was expressed in hEx45KI mice, but no expression was detected in hEx45KI-mdx44 mice (Fig. 3d). Histological H&E analysis showed necrosis, centrally nucleated fibers, and the loss of membrane-associated dystrophin protein in hEx45KI-mdx44, but not in hEx45KI mice (Fig. 3e). hEx45KI-mdx44 mice also showed calf hypertrophy and greater weight of the GC than C57BL/6J or hEx45KI mice at 9- and 13-week of age (Fig. 3f). Finally, the plasma CK level in hEx45KI-mdx44 mice was significantly higher than that in C57BL/6J or hEx45KI mice (Fig. 3g).

**Dual sgRNA delivery restored dystrophin expression.** To evaluate the exon skipping efficiency by the sgRNAs that target human exon 45 in hEx45KI-mdx44 mice, we tested the combination of two sgRNAs. LNP-hEx45 sgRNA #1, LNP-hEx45 sgRNA #23 or both were intramuscularly administered with LNP-Cas9 mRNA into the tibialis anterior muscle (TA) of hEx45KI-mdx44 mice. TA samples were collected a week after administration and measured for the skipping efficiency of exon 45 by qRT-PCR. Similar to the results in DMD-iPSC-derived myoblasts (Fig. 2d), the skipping efficiencies induced by sgRNA#1 were higher than sgRNA#23, and co-injection of sgRNA#1 and #23 increased exon skipping efficiency in mouse compared with a single sgRNA (Fig. 4a).

In addition, we compared the efficacy of our TCL053-based LNP with DLin-MC3-DMA (MC3)-based LNP, which is a major LNP formulation of the first RNAi therapeutic Onpattro. We prepared TCL053-LNP-CRISPR and MC3-LNP-CRISPR encapsulating Cas9 mRNA, hEx45 sgRNA#1, and #23 with several lipid compositions including a similar composition to ours or Onpattro (Supplementary Fig. 6a). Then, these LNP-CRISPR were intramuscularly injected into the TA of hEx45KI-mdx44 mice. The result showed that the genome editing and exon skipping efficacy of TCL053-LNP-CRISPR were higher than those of MC3-LNP-CRISPR in all compositions (Supplementary Fig. 6b, c). We also compared the efficacy of TCL053-LNP with commercially available in vivo mRNA delivery reagent, in vivo-jetRNA, and found that exon skipping efficacy of TCL053-LNP-CRISPR was higher than that of in vivo-jetRNA (Supplementary Fig. 6d).

**Long-term, stable induction of exon skipping and dystrophin protein.** We next examined the period of efficacies induced by the genome editing. LNP-CRISPR (gRNA#1 + #23) and phosphorodiamidate morpholino oligomer-based ASO targeting human exon 45 were intramuscularly injected into the TA and GC of hEx45KI-mdx44 mice, and the muscles were collected 0.5, 1, 3, 6, or 12 months after the injection. The exon skipping efficiency was analyzed by qRT-PCR in GC, and the dystrophin recovery rate was detected by Western blotting in TA. The exon skipping and dystrophin expression in TA and GC of ASO-injected mice were promptly and transiently increased and then gradually decreased (Fig. 4b, c). On the other hand, the levels of exon skipping efficiency in GC of LNP-CRISPR-injected mice were stable for a year, and the dystrophin expression in TA maintained (Fig. 4b, d).

To assess any tissue damage, we performed histological analysis of the injected TA. Seven days after the single administration of LNP-CRISPR, muscle histology was examined by H&E staining. A mild infiltration of immune cells and fibrotic appearance were observed in the local area of the injection in both PBS- and LNP-treated samples (Supplementary Fig. 7a). We also collected blood samples from the treated mice and observed the GLDH level as a marker of liver injury in DMD model[38]. The LNP-CRISPR-injection group showed no significant elevation compared with the PBS-injection group, and the value returned to basal level at 7 days post injection (Supplementary Fig. 7b). Similarly, we assessed several cytokines in plasma. Some cytokines (MCP-1, MIP-1β, TNF-α, and IFN-γ) showed a transient elevation at either 6 or 24 h post injection, although all returned to basal level at 7 days post injection (Supplementary Fig. 7c). These results indicate that the intramuscular injection of LNP might trigger some cytokine release, yet all blood biomarkers were transient, lasting less than one week.

**Low immunogenicity of LNP allows repeated administration.** Given the transient and mild nature of the LNP-related inflammation, we tested whether repeated intramuscular injections of LNP-CRISPR are feasible. To monitor exon 45 skipping non-invasively over time, we used Luc reporter mice that were previously generated by the insertion of Luc gene divided by human *DMD* exon 45[35] (Fig. 5a). We first injected LNP-CRISPR or AAV-CRISPR into the left leg of the reporter mice, and 28 days later we injected the same dose of LNP-CRISPR or AAV-CRISPR into the other (right) leg in the same reporter mice. 28 days after the second injection, bioluminescence images were obtained to detect genomic exon skipping. Excitingly, Luc expression was observed in both legs of reporter mice treated with LNP-CRISPR (Fig. 5b, c). In contrast, bioluminescence was detected only in the left leg after the first AAV-CRISPR injection, but not in the right leg after the second injection regardless of the AAV dose.

Next, we examined whether repeated administration leads to a cumulative therapeutic effect. We injected LNP-CRISPR into the TA of hEx45KI-mdx44 mice one, two, or three times with one-month intervals between injections. TAs were collected 2 weeks after the final injection, and the level of dystrophin was analyzed by Western blotting. The recovery rate of dystrophin was increased dose-dependently (Fig. 5d). We also tried up to 6-time injections of LNP-CRISPR with a single sgRNA (#1) into hEx45KI-mdx44 mice. The genome editing efficacy was increased with the number of doses (Supplementary Fig. 8a), and the mice injected 6 times showed exon skipping at the RNA level (Supplementary Fig. 8b, c) as well as recovery of dystrophin protein expression (Supplementary Fig. 8d). Dystrophin protein recovery was better at 40 days post treatment than that at 4 days post treatment.

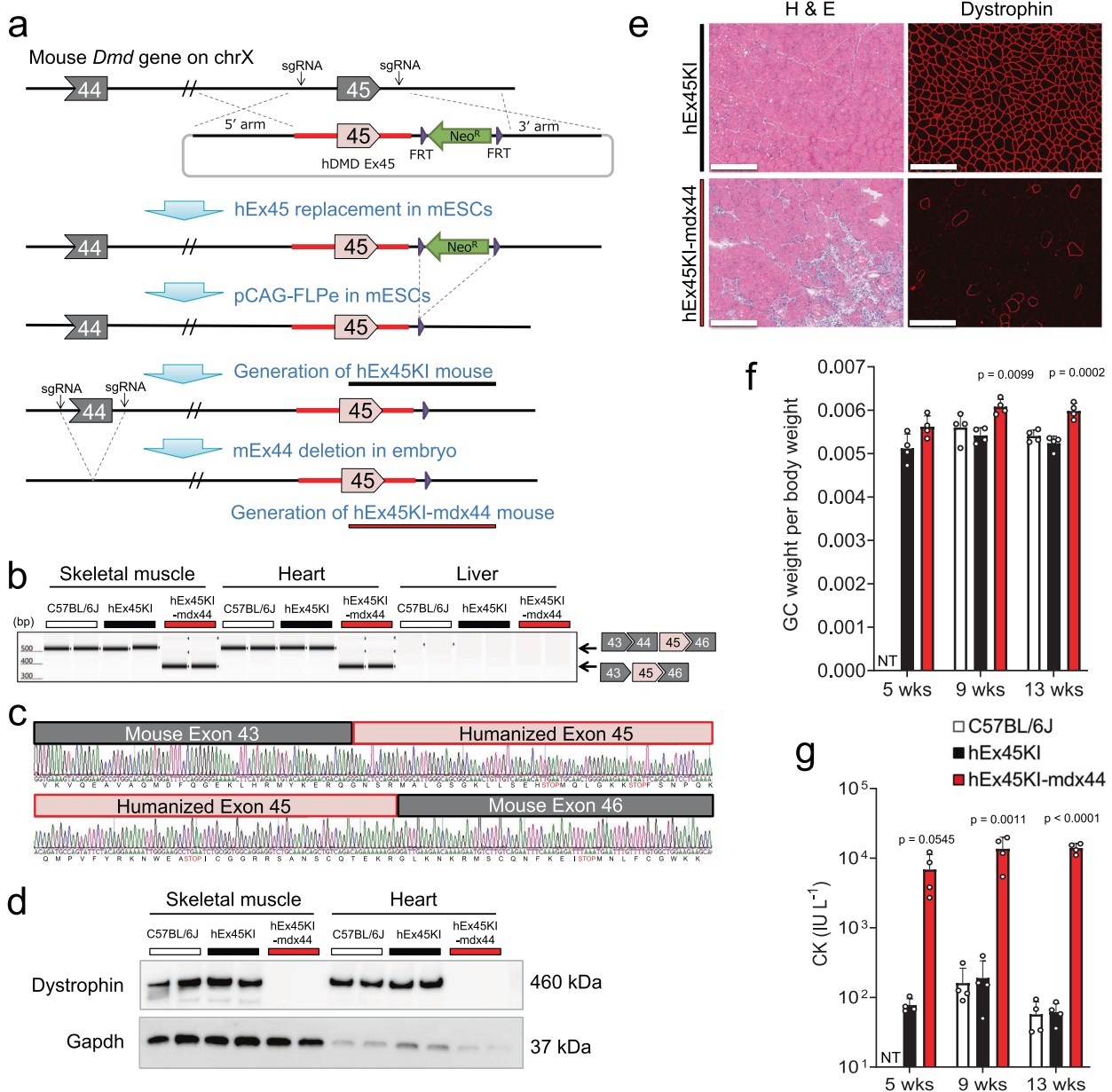

**Fig. 3 Generation of an exon 45 humanized mouse model to assess sgRNAs that target human dystrophin. a** Schematic illustration showing the generation of humanized *DMD* exon45 KI-Dmd exon44 knock-out mice (hEx45KI-mdx44). mESCs mouse embryonic stem cells. **b** Confirmation of exon 44 (ΔEx44) deletion was assessed by RT-PCR in GC, heart, and liver from C57BL/6J, humanized *DMD* exon45 KI (hEx45KI) or hEx45KI-mdx44 mice. Upper band shows normal transcript and lower band indicates ΔEx44. $n = 2$ mice per each strain. Similar results were observed in more than 3 experiments. **c** The lower band from **b** was sequenced to confirm exon–exon junctions. The black line shows mouse *Dmd*, and red line shows humanized *DMD* sequences. **d** Lack of dystrophin protein expression in GC and heart were confirmed by Western blot analysis. $n = 2$ mice per each strain. Lack of dystrophin protein in hEx45KI-mdx44 mice was confirmed in more than 3 experiments. **e** Hematoxylin and eosin (H & E) staining and immunohistochemical staining of dystrophin in TA of hEx45KI (upper) and hEx45KI-mdx44 (lower) mice 9 weeks of age. $n = 4$ mice per each strain, but only a representative image from each strain is shown. Scale bars indicate 50 μm. **f, g** GC weight per body weight (no unit) (**f**) or plasma creatine kinase (CK) level (IU L$^{-1}$, international unit per liter) (**g**). White, black, and red columns show C57BL/6J, hEx45KI, and hEx45KI-mdx44 mice, respectively. NT not tested. Data are represented as means ± S.D. ($n = 4$ mice). $p$ values by two-sided Dunnett's test showed significantly different from same-age hEx45KI mice.

Finally, we injected LNP-CRISPR with dual sgRNAs (#1 and #23) into the TA of hEx45KI-mdx44 mice 6 times over a 2-week period. We collected tissue sections of the injected muscles 2 months after the final injection and counted the number of muscle fibers with dystrophin expression and central nuclei. We observed 38.5% of fibers were dystrophin-positive with LNP-CRISPR by immunohistochemistry (Fig. 5e, f). The rate of muscle fibers with a central nucleus, which is a hallmark of damaged and regenerating myofibers, was significantly decreased compared with sham (PBS) treatment (Fig. 5g).

To investigate the immunogenicity of LNP or AAV, we injected LNP-CRISPR or AAV-CRISPR multiple times and measured the level of anti-Cas9 antibody at various time points (Supplementary Fig. 9a). AAV-CRISPR showed an elevated production of anti-Cas9 antibody in mouse serum over time, whereas the antibody levels were much less in LNP-CRISPR-

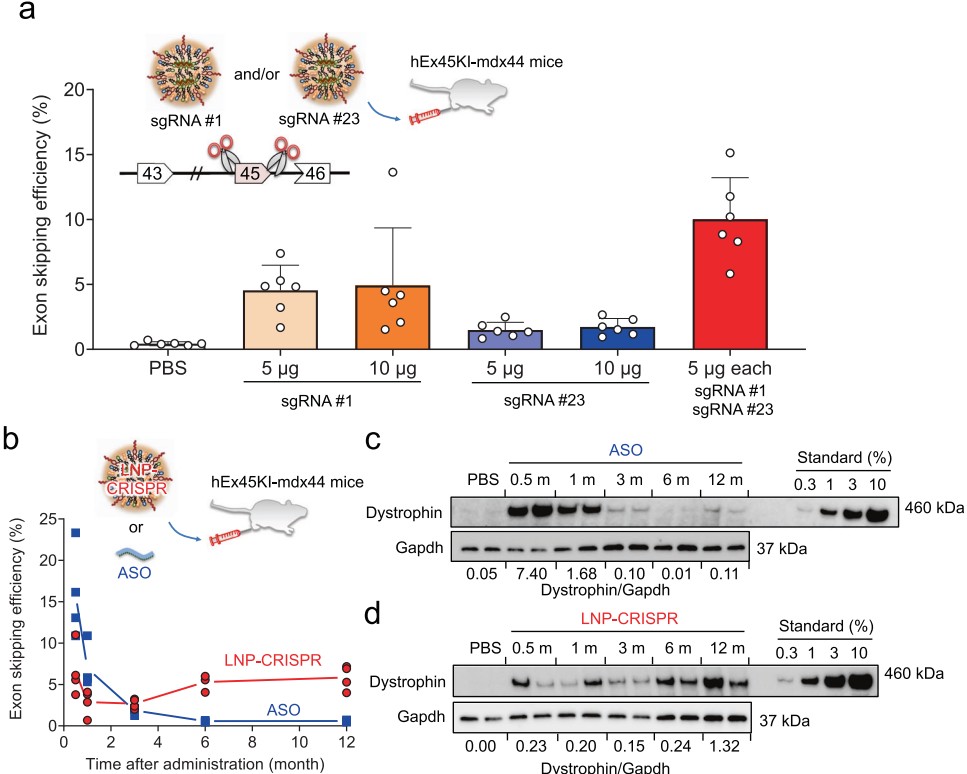

**Fig. 4 LNP delivery of dual sgRNAs into humanized model mice restored dystrophin protein over 12 months. a** Effect of dual sgRNA combination on exon skipping efficiency. A week after the single administration of LNPs encapsulating 10 μg of Cas9 mRNA and the indicated amounts of sgRNA(s) and mRNA from TA in hEx45KI-mdx44 mice (*n* = 6 mice) was assessed by qRT-PCR. Data are represented as means ± S.D. **b** Long-term effect of exon skipping was measured by qRT-PCR in GC after a single injection of ASO (phosphorodiamidate morpholino oligomer) or LNP-CRISPR (sgRNA #1 + #23) into hEx45KI-mdx44 mice. Each dot represents data from an individual mouse, and average values from each time point (*n* = 4 mice) are connected by a solid line to show the trend of each group over time. **c**, **d** The recovered dystrophin level was assessed by Western blotting in TA after a single intramuscular administration of ASO (**c**) or LNP-CRISPR (**d**). Numbers below the WB images shows relative amount of dystrophin expression normalized by Gapdh amount at each time point. *n* = 4 mice per each time point, but only two mice of Western blot images are shown.

injected mice (Supplementary Fig. 9b), suggesting the low immunogenicity of LNP delivery even with multiple injections compared with AAV delivery.

**Widespread delivery of LNP-CRISPR by intravenous limb perfusion.** Since body movement is controlled by multiple muscle groups, it will be important to treat as many muscles as possible by a single treatment. To target a wide range of locoregional skeletal muscle tissues with LNP-CRISPR, we thought to investigate limb vein injection with local restriction. For this, we placed a tourniquet at the base of the quadriceps and injected LNP-CRISPR into the dorsal saphenous vein of hEx45KI-mdx44 mice (Fig. 6a). To assess the effect of the injection volume, we injected various volumes of LNP-CRISPR but a consistent total RNA dose and injection speed. One week after the injection, exon skipping efficiency was evaluated in various muscle groups of the hindlimb. When the injection volume of the limb perfusion was greater than 5 mL/kg, exon skipping was detected in various limb muscles except for the quadriceps where the tourniquet was placed. In contrast, when LNP-CRISPR was intramuscularly injected into TA, exon skipping was detected mainly in TA and weakly in extensor digitorum longus muscle (Fig. 6b). Next, we escalated the amount of LNP and RNA from 1 mg/kg to 10 mg/kg (Cas9 mRNA: sgRNA #1: sgRNA #23 = 2: 1: 1, w/w/w) while keeping the injection volume fixed at 5 mL/kg. As expected, the exon skipping efficiency of LNP-CRISPR by limb perfusion was dose-dependent, and the overall exon skipping was enhanced at a high dose (10 mg/kg as total RNA) (Fig. 6c). In addition, we

observed dose-dependent dystrophin restoration in both GC and TA by the single limp perfusion of LNP-CRISPR (Fig. 6d).

We also measured the plasma GLDH level following the limb perfusion of LNP-CRISPR to assess the hepatotoxicity. Plasma GLDH level increased immediately after the limb perfusion in all the treatment groups, and remained higher in the 10 mg/kg group than in the PBS group for several days (Supplementary Fig. 10a). Since PBS-injected group also showed similar trend of plasma GLDH level as LNP-CRISPR injected group, limb perfusion procedure itself may have some liver damage. Anyhow, all the treatment groups returned to basal level after 7 days of injection (Supplementary Fig. 10b), indicating a transient nature of liver damage by limb perfusion approach.

## Discussion
Here, we showed that the transient delivery of Cas9 mRNA and sgRNA by a novel formulation of LNP could induce genome editing in skeletal muscle tissue in mice. DMD is a systematic disorder, hence systematic delivery of CRISPR-Cas9/sgRNA is preferable for treatment[39]. However, intravenous injection raises the risk of inducing genome editing in off-target tissues, such as the liver, blood vesicles, or germ cells. Although male germ cells are protected by the blood–testis barrier, the genomic modification of germ cells is prohibited by gene therapy regulations from an ethical point of view[40]. To this end, we investigated intramuscular injection, which is a standard injection route for many existing medical and myoblast transplantations. Unquestionably, it is challenging to cover all skeletal muscle tissues by a single treatment, because skeletal muscle

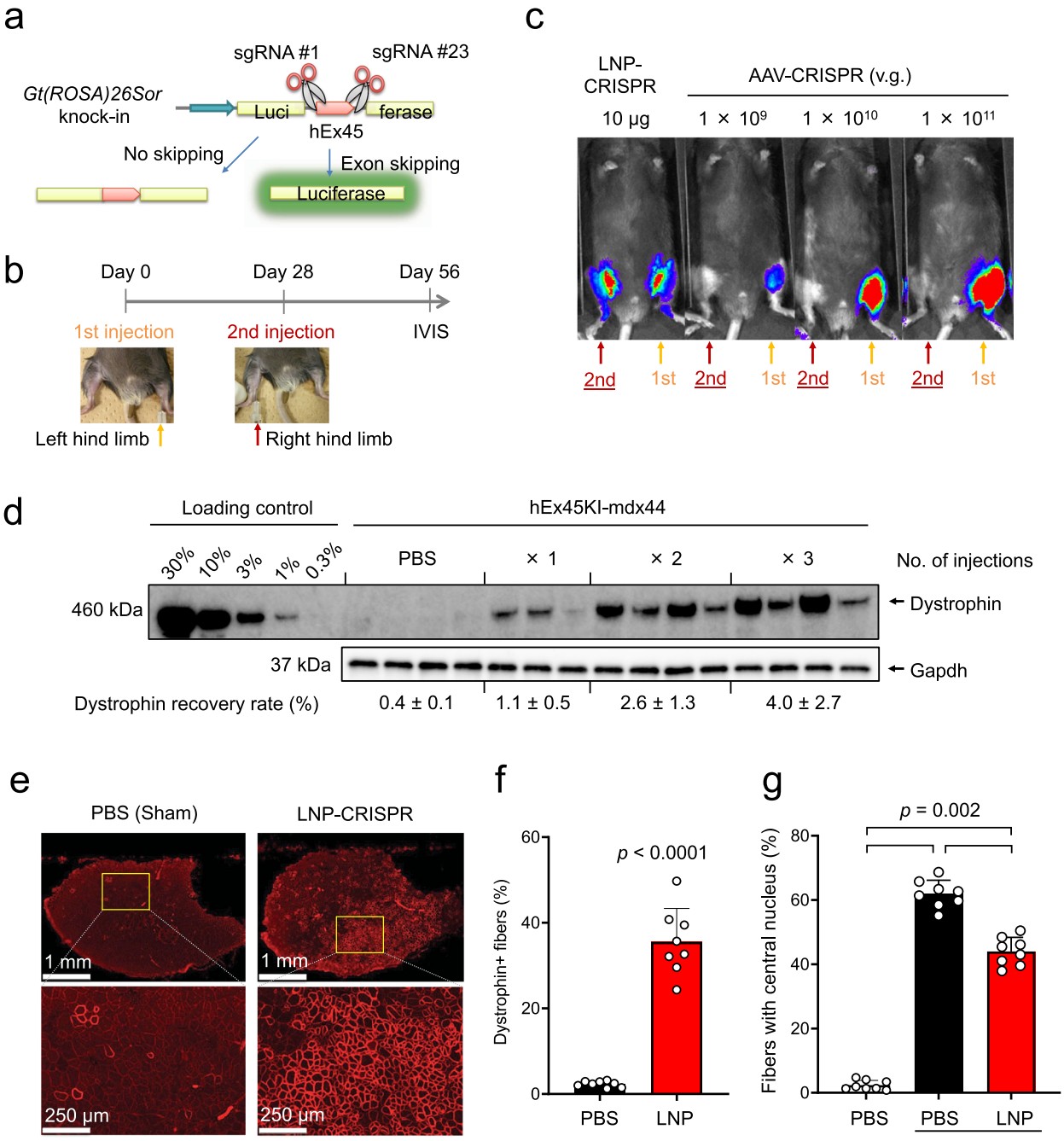

**Fig. 5 LNP-Cas9/sgRNA delivery enables repeated administrations and persistent recovery of dystrophin protein. a** Schematic illustration of the detection of exon skipping efficacy in CAG-Luc2 hDMDEx45 KI reporter mice. Luciferase expression can be detected only if human dystrophin exon 45 skipping occurs. **b** LNP-CRISPR (5 µg of each sgRNA and 10 µg Cas9 mRNA) or AAV-CRISPR ($1 \times 10^9$, $1 \times 10^{10}$, or $1 \times 10^{11}$ v.g.) was injected into the left leg of CAG-Luc2 hDMDEx45 KI mice first. Then after 28 days, LNP-CRISPR or AAV-CRISPR was injected into the right leg of the same mice at the same dose as the first injection. Another 28 days later, exon skipping was detected as luciferase activity by IVIS. **c** Representative bioluminescence images of CAG-Luc2 hDMDEx45 KI mice. LNP-CRISPR successfully induced exon skipping in the right leg after the secondary injection, whereas AAV-CRISPR failed to induce. **d** Administration frequency-dependent dystrophin recovery in TA one week after one, two, or three intramuscular injections of LNP with one-month intervals in hEx45KI-mdx44 mice. Data are represented as means ± S.D. ($n = 3$ mice for one injection group, $n = 4$ mice for other groups).
**e** Immunohistochemical staining of dystrophin 8 weeks after 6 injections of LNP-CRISPR (5 µg of each sgRNA and 10 µg Cas9 mRNA, 6 shots in 2 weeks). Representative images in each group of 8 mice are shown. **f** Percentage of dystrophin positive fibers in TA. Data are represented as means ± S.D. ($n = 8$ mice). $p$ values by two-sided Aspin–Welch's $t$-test showed significantly different from hEx45KI mice. **g** Percentage of fibers with a central nucleus in TA. Data are represented as means ± S.D. ($n = 8$ mice). A significant difference was found among the three groups (two-sided Steel-Dwass test).

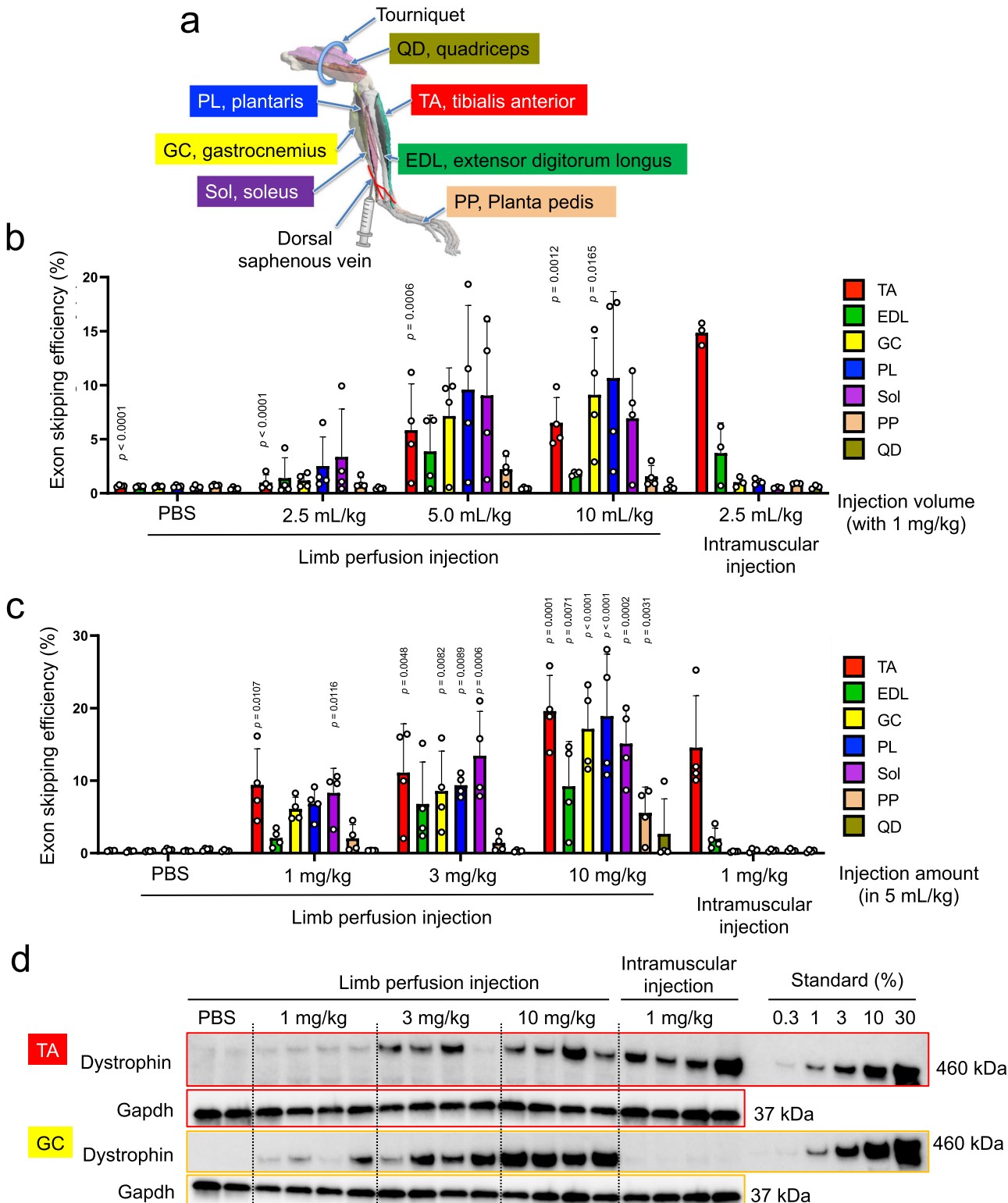

**Fig. 6 Intravenous delivery of LNP-Cas9/sgRNA by limb perfusion. a** Schematic illustration of mouse hindlimb and relevant muscle groups. The 3D model of hindlimb is adapted from ref. [50]. LNP-CRISPR was injected into the dorsal saphenous vein. **b** Effect of the injection volume (2.5–10 mL/kg) on exon skipping efficiency in each muscle group of the lower hindlimb following limb perfusion of LNP-CRISPR in hEx45-mdx44 mice. TA tibialis anterior, EDL extensor digitorum longus, GC gastrocnemius, PL plantaris, Sol soleus, PP planta pedis, QD quadriceps. Data are represented as means ± S.D. (n = 4 mice for limb perfusion injection groups, n = 3 mice for intramuscular injection group). p value by two-sided Dunnett's test showed significantly different from intramuscular injection (2.5 mL/kg) in the same skeletal muscle. **c** Effect of the injection dose (1–10 mg/kg) of total RNA packaged in LNP under 5 mL/kg injection volume. Data are represented as means ± S.D. (n = 4 mice). p value by two-sided Williams' test showed significantly different from limb perfusion-PBS of the same skeletal muscle. **d** Dose-dependent dystrophin restoration in TA and GC following 14 days of limb perfusion of LNP-CRISPR (n = 4 mice for each LNP injection group, n = 2 mice for PBS injection group).

consists of a large portion (~40%) of body and each muscle is separated by fascia. Because of this, repeated administration is desirable to incrementally cover the treatable muscles. Furthermore, we investigated the limb perfusion injection method, as it is a scalable and regionally restricted delivery approach, yet it can target multiple skeletal muscle groups. Hydrodynamic injection with a large volume is a major approach for delivering DNA vectors[41]; however, our injection volume (for the 5 mL/kg condition, 125 μL per leg per 25 g of mouse) is much smaller than hydrodynamic limb vein injection methods (i.e., 1 mL per leg[42]), presumably due to the high delivery activity of LNP. The small volume would be advantageous when performing limb perfusion in patients to avoid any compartment syndrome[41].

Several reports have used AAV vector delivery, but genomic integration of vector DNA fragments is a concern[12,15]. In addition, as shown previously and in Fig. 5c, the same serotype of AAV vector cannot be administrated repeatedly[16]. Hence, the feasibility of repeated administration with our LNP is a major advantage as a delivery vehicle of the CRISPR-Cas9 system.

ASO is a pioneering technology to induce therapeutic exon skipping, and in fact, there are a couple of ASO drugs already been approved for clinical use[2]. Molecular size of ASO is much smaller than LNP, and ASO can be administrated via intravenous injection to target body-wide skeletal muscle. However, patients require life-long administration with ASO therapy, and once the treatment is terminated, the therapeutic effect will be reverted. Although Cas9 protein expression from LNP is transient and disappeared within a few days (Supplementary Fig. 3b, c), the exon skipping effect and dystrophin protein expression can be maintained for several months or more, so long as genome edited cells survive. This can be another advantage of LNP-CRISPR approach over ASO treatment (Fig. 4b, d).

Because of the sequence-specific nature of the CRISPR-Cas9 genome editing product, CRISPR-sgRNA targeting human sequences to treat patients cannot be tested in animals directly unless the targeted sequence is conserved or the animal sequence is replaced with the human sequence[8]. Since we optimized our sgRNA targets for efficient exon skipping by using patient-derived iPS cells previously[43], here we generated a humanized mouse model (hEx45KI-mdx44) and tested our human *DMD* gene-targeting sgRNAs. Although the muscle phenotype is somewhat mild, which is similar to mdx mouse[44], genome editing and exon skipping efficiencies can be assessed with our humanized mouse.

In the clinical application of CRISPR-Cas9, off-target mutagenesis is a major safety concern[45,46]. However, the site and degree of off-target mutagenesis in the mouse genomic context may not reflect the risk in the human genome. For this reason, we first performed CIRCLE-seq off-target analysis with human genomic DNA to list potential biochemical cleavage sites and subsequently assessed the risk of mutagenesis in cultured human cells after the prolonged overexpression of Cas9/sgRNA from a *piggyBac* vector. Although several candidate DNA cleavage sites can be identified by CIRCLE-seq experiments, none of the candidate sites we tested induced mutagenesis in the cells.

In conclusion, we have developed a chemically defined LNP-delivery system which enables the transient delivery of CRISPR-Cas9 mRNA and sgRNA into skeletal muscle tissues in vivo. Although further development and assessments will be required to ensure the efficacy and safety, this is an important step towards the development of genome editing therapy for muscular dystrophy and beyond.

## Methods

### Chemical synthesis of 3-((4-(Dimethylamino)butanoyl)oxy)-2,2-bis(((9Z)-tetradec-9-enoyloxy) methyl) propyl (9Z)-tetradec-9-enoate (TCL053). To a mixture of 2,2-bis(hydroxymethyl)propane-1,3-diol (5.45 g), 1H-imidazole (2.72 g)

and DMF (190 mL), a solution of tert-butylchlorodimethylsilane (3.01 g) in DMF (10 mL) was added at room temperature. After stirring for 24 h, the reaction mixture was concentrated under reduced pressure. The residue was diluted with ethyl acetate, washed three times with water and once with saturated brine, and then dried over anhydrous sodium sulfate, and the solvent was distilled off under reduced pressure. The residue was purified by silica gel column chromatography (ethyl acetate/hexane) to afford the 2-(((tert-Butyldimethylsilyl)oxy)methyl)-2-(hydroxymethyl)propane-1,3-diol compound (2.25 g).

[1]H NMR (300 MHz, CDCl$_3$) δ ppm 0.08 (6H, s), 0.90 (9H, s), 2.53 (3H, t, J = 5.5 Hz), 3.66 (2H, s), 3.73 (6H, d, J = 5.5 Hz)

Next, to a solution of 2-(((tert-butyldimethylsilyl)oxy)methyl)-2-(hydroxymethyl) propane-1,3-diol (258 mg), (9Z)-tetradec-9-enoic acid (769 mg) and DMAP (126 mg) in DMF (3 mL), 1-ethyl-3-(3-dimethylaminopropyl) carbodiimide hydrochloride (790 mg) was added at room temperature. After stirring for 18 h, the reaction mixture was diluted with ethyl acetate, washed twice with water and once with saturated brine, and then dried over anhydrous sodium sulfate, and the solvent was distilled off under reduced pressure. The residue was purified by silica gel column chromatography (NH, ethyl acetate / hexane) to afford the 3-((tert-Butyl(dimethyl)silyl)oxy)−2,2-bis(((9Z)-tetradec-9-enoyloxy)methyl) propyl (9Z)-tetradec-9-enoate compound (860 mg).

[1]H NMR (300 MHz, CDCl$_3$) δ ppm 0.03 (6H, s), 0.81-0.96 (18H, m), 1.18-1.41 (36H, m), 1.53-1.67 (6H,m), 1.91-2.10 (12H, m), 2.29 (6H, t, J = 7.6 Hz), 3.58 (2H, s), 4.08 (6H, s), 5.27-5.43 (6H, m)

Third, to a solution of 3-((tert-butyl(dimethyl)silyl)oxy)−2,2-bis(((9Z)-tetradec-9-enoyloxy) methyl)propyl (9Z)-tetradec-9-enoate (5.91 g) in THF (120 mL), a mixture of a THF solution of TBAF (1 M, 14.85 mL) and acetic acid (4.91 mL) was added at room temperature. After stirring for 3 days, the reaction mixture was concentrated under reduced pressure. The residue was diluted with ethyl acetate, washed once with saturated aqueous solution of sodium hydrogen carbonate and once with saturated brine, and then dried over anhydrous sodium sulfate, and the solvent was distilled off under reduced pressure. The residue was purified by silica gel column chromatography (ethyl acetate/hexane) to afford the 3-Hydroxy-2,2-bis(((9Z)-tetradec-9-enoyloxy)methyl)propyl(9Z)-tetradec-9-enoate compound (4.96 g).

[1]H NMR (300 MHz, CDCl$_3$) δ ppm 0.82–0.97 (9H, m), 1.16–1.42 (36H, m), 1.52–1.68 (6H, m), 1.90–2.12 (12H, m), 2.32 (6H, t, J = 7.6 Hz), 2.52 (1H, t, J = 7.0 Hz), 3.49 (2H, d, J = 7.0 Hz), 4.11 (6H, s), 5.26-5.42 (6H, m)

Finally, to a solution of 3-hydroxy-2,2-bis(((9Z)-tetradec-9-enoyloxy)methyl) propyl (9Z)-tetradec-9-enoate (4.96 g) and 4-(dimethylamino) butanoic acid hydrochloride (2.19 g) in DMF (20 mL), 1-ethyl-3-(3-dimethyl aminopropyl) carbodiimide hydrochloride (2.50 g) was added at room temperature. After stirring for 18 h, the reaction mixture was diluted with ethyl acetate, washed once with saturated aqueous solution of sodium hydrogen carbonate and once with saturated brine, and then dried over anhydrous sodium sulfate, and the solvent was distilled off under reduced pressure. The residue was purified by silica gel column chromatography (NH, ethyl acetate/hexane) to afford the 3-((4-(Dimethylamino)butanoyl)oxy)−2,2-bis(((9Z)-tetradec-9-enoyloxy) methyl) propyl (9Z)-tetradec-9-enoate (TCL053) compound (5.31 g).

[1]H NMR (300 MHz, CDCl$_3$) δ ppm 0.82-0.94 (9H, m), 1.20-1.42 (36H, m), 1.50-1.66 (6H, m), 1.69-1.83 (2H, m), 1.90-2.10 (12H, m), 2.20 (6H, s), 2.23-2.41 (10H, m), 4.11 (8H, s), 5.23-5.44 (6H, m), MS m/z (M + H): 874.75

**LNP formulation.** TCL053 and TCL065 ionizable lipids were synthesized and obtained from Drug Discovery Chemistry Labs in Takeda Pharmaceutical Company, Ltd. DLin-MC3-DMA ionizable lipid was purchased from MedChemExpress (Monmouth Junction, NJ). Structural lipids DPPC (1,2-dipalmitoyl-sn-glycero-3-phosphocholine), DOPE (1,2-dioleoyl-sn-glycero-3-phosphoethanolamine), DSPC (1,2-distearoyl-sn-glycero-3-phosphocholine), and DMG-PEG (1,2-dimyristoyl-sn-glycero-3-methoxypolyethyleneglycol 2000) were purchased from NOF Corporation (Tokyo, Japan), and cholesterol from Avanti Polar Lipids (Alabaster, AL). Firefly luciferase (luc) mRNA and Cas9 mRNA (CleanCap® Cas9 mRNA (5moU), uridines were substituted with 5-methoxyuridine, catalog no. L-7206, 4521-mer) were purchased from Trilink Biotechnologies (San Diego, CA). Chemically modified sgRNAs (2'OMe and phosphorothioate modification of three nucleotides at the 5′ and 3′ end) were synthesized by GeneDesign (Osaka, Japan). Lipid components (60% ionizable lipid, 10.6% structural lipid, 27.3–28.7% cholesterol, 0.7–2.1% DMG-PEG, in mol%) were dissolved in 90% ethanol, and RNA (mRNA only, sgRNA only, or a mixture of both) was dissolved in 10 mM MES (2-(N-morpholino) ethanesulfonic acid) buffer (pH 5.5). For MC3-LNP-IV, lipid components (50% Dlin-MC3-DMA, 10% DSPC, 28.5% cholesterol, 1.5% DMG-PEG, in mol%) were dissolved in ethanol, and RNAs (Cas9 mRNA, hEx45 sgRNA #1, and hEx45 sgRNA #23) were dissolved in 50 mM sodium citrate (pH 4.0).

LNP-mRNA and LNP-sgRNA were prepared by mixing the organic lipid components and the aqueous RNA solution at a lipid:RNA ratio of 23:1 (w/w), with a microfluidic laminal flow rate of 8 mL/min (aqueous:organic flow rate ratio of 2:1) using a NanoAssemblr Benchtop (Precision Nanosystems, Vancouver, Canada). Then, the mixture was dialyzed for 1 h at 4 °C against RNase-free water followed by dialysis for 18 h at 4 °C against PBS (pH 7.4) using a Slide-A-Lyzer Dialysis Cassette 20 K MWCO (Thermo Fisher Scientific, Waltham, MA). Resultant LNP was concentrated with an Amicon Ultra 30 K MWCO (Merk

**Table 1 Primers used for detecting exon skipping.**

| | Unskipped | Skipped |
|---|---|---|
| Standard DNA | 5′-AGCAATCCTCAAAAACAGATGCCAGTAT TCTACAGGAAAAATTGGGAAGCCTGAATCT GCGGTGGCAGGAGGTCT-3′ | 5′-GAAGCCGTGGCACAGATGGATTTCCAG GGGGAAAAACTTCATAGAATGTACAAGGA ACGACAAGGGATTGAAGAACAAAAGAATGTCT-3′ |
| Forward primer | 5′-TCCTCAAAAACAGATGCCAGTATTC-3′ | 5′-CGTGGCACAGATGGATTTCC-3′ |
| Reverse primer | 5′-TCCTGCCACCGCAGATTC-3′ | 5′-TTCTTTTGTTCTTCAATCCCTTGTC-3′ |
| FAM-probe | 5′-ACAGGAAAAATTGGGAAGC-3′ | 5′-ACTTCATAGAATGTACAAGGAA-3′ |

Millipore, Burlington, MA) and filtrated through a 0.2-μm syringe filter (AGC Techno Glass, Shizuoka, Japan). The LNP size was determined by dynamic light scattering using a Zetasizer Nano ZS (Malvern Panalytical, Worcestershire, UK), and the RNA packaging efficiency was measured by a Quant-it RiboGreen RNA Assay Kit (Thermo Fisher Scientific), as manufacture's instruction[47].

**AAV preparation**. AAV2-luc was purchased from Cell Biolabs (San Diego, CA). AAV vectors containing hEx45 sgRNAs #1 and #23 were constructed by cloning synthetic oligos (Eurofins Genomics, Tokyo, Japan) into pAAV-Guide-it-Down vector (Takara Bio, Shiga, Japan). pAAV-Guide-it Up vector was purchased from Takara Bio. These vectors were packaged into AAV-DJ (Cell Biolabs) using AAVpro293 cells (Takara Bio) and purified using an AAVpro Purification Kit (Takara Bio).

**Myogenic differentiation of DMD-iPS cells and LNP-CRISPR transfection.** Dox-inducible MYOD1-expressing DMD patient iPS (DMD-iPS) cells (clone CiRA00111, available from the RIKEN BioResource Research Center as HPS0383) were previously generated[48]. The DMD-iPS cells were seeded on a Matrigel-coated plate and cultured in StemFit AK02N (Ajinomoto, Tokyo, Japan) containing 10 μM Y-27632 dihydrochloride (Tocris Bioscience, Bristol, UK). The next day, medium was changed with fresh ReproCell Primate ES Cell medium (REPROCELL, Kanagawa, Japan). After 24 h, the medium was changed to ReproCell Primate ES Cell medium supplemented with 1 μg/mL doxycycline (DOX) (Sigma-Aldrich, Saint Louis, MO). Twenty-four hours after the addition of DOX, the medium was changed to α-MEM (Sigma-Aldrich) containing 5% KnockOut Serum Replacement (Thermo Fisher Scientific) and 1 μg/mL DOX. After 3 days, LNP-Cas9 mRNA (1 μg mRNA) was transfected with LNP-hEx45 sgRNA #1 (1 μg sgRNA), LNP-hEx45 sgRNA #23 (1 μg sgRNA) or LNP-hEx45 sgRNA #1 + #23 (0.5 μg each RNA) into myoblast cells derived from DMD-iPS cells, and the cells were incubated at 37 °C for 3 days.

**Generation of humanized DMD model mice and animal experiments.** Human DMD exon 45 knock-in (hEx45KI) mice were generated by replacing mouse exon 45 and the flanking intronic sequence (total 1.5 kb in size) with the corresponding human exon 45 sequence by CRISPR-Cas9/sgRNA-mediated targeting, and hEx45KI-Dmd exon 44 knock-out (hEx45KI-mdx44) mice were generated by deleting mouse exon 44 with CRISPR-Cas9 dual sgRNAs (performed by Axcelead, Kanagawa, Japan). CAG-Luc2 hDMDEx45 KI mice were previously generated by targeting the Gt(ROSA)26Sor locus with a pCAGGS-Luc2-hEx45 reporter construct using the CRISPR-Cas9 system[35]. C57BL/6J mice were purchased from CLEA Japan (Kanagawa, Japan). All mice were housed under SPF condition with free-food and water supply with 12 h dark/light cycle, controlled temperature (around 23 °C), and controlled humidity (around 55%). Mice were anesthetized with isoflurane and intramuscularly injected with LNP, AAV or phosphor-odiamidate morpholino oligomer (5′-GCTGCCCAATGCCATCCTG-GAGTTCCTG-3′, obtained from Gene Tools, Philomath, OR) into the gastrocnemius (GC) or tibialis anterior (TA) muscle. We used a 29-gage (G) needle and inserted 5–6 mm to inject 50 μL (for TA, IHC analysis) or 100 μL (for GC, IVIS analysis) of the sample solution at a speed of 12–25 μL per second. For repeated injections, we tried to inject at the same site each time. For limb perfusion, a tourniquet was placed on the proximal part of the hindlimb before the injection[49]. LNP in a volume of 2.5–10 mL/kg was injected into the dorsal saphenous vein in approximately 10 s with a 32-G needle. Five minutes after the injection, the tourniquet was removed. All in vivo experiments were evaluated and approved by the Institutional Animal Care and Use Committee at Takeda Pharmaceutical Company, Ltd.

**Detection of luciferase bioluminescence.** Mice were anesthetized with isoflurane and intraperitoneally administered 3 mg D-Luciferin (Promega, Madison, WI) that was dissolved in PBS. Ten minutes after the injection of D-Luciferin, images were captured using IVIS. Hairs around the GC and abdomen of mice were shaved prior to imaging to increase the sensitivity of the luminescence detection.

**Measurement of exon skipping efficiency.** Total RNA of DMD-iPS cell-derived myoblast cells were extracted by RIPA buffer (FUJIFILM Wako Pure Chemical,

Osaka, Japan) and an miRNeasy Mini Kit (Qiagen, Hilden, Germany) according to the manufacturer's instructions. Extracted total RNA was reverse transcribed into cDNA using a high capacity-RNA-to-cDNA Kit (Thermo Fisher Scientific) according to the manufacturer's instructions and PCR amplified using PrimeSTAR GXL DNA polymerase (Takara Bio) and a primer set. The PCR product was purified by a QIAquick PCR Purification Kit (Qiagen) and analyzed by High Sensitivity D1000 Screen Tape using an Agilent 4200 TapeStation (Agilent Technologies, Santa Clara, CA).

The total RNA in mouse TA or GC was extracted by Qiazol (Qiagen) and an RNeasy 96 Kit (Qiagen) according to the manufacturer's instructions. One microgram of total RNA was reverse transcribed into cDNA using a high capacity-RNA-to-cDNA Kit according to the manufacturer's instructions. Standard DNAs for skipped or unskipped products or 100 ng of cDNA products were PCR amplified by FastStart Universal Probe Master (Roche, Basel, Switzerland) and appropriate primer/probe sets (Table 1). The amount of each PCR product was calculated by using standard DNA, and the exon skipping efficiency was determined by the following formula: $100 \times$ amount of skipped product/(amount of skipped product + amount of unskipped product).

**Dystrophin protein detection by Western blotting.** Harvested cells or collected tissues were mixed with RIPA buffer (FUJIFILM Wako Pure Chemical) containing protease inhibitor cocktail (Sigma-Aldrich) and EDTA (FUJIFILM Wako Pure Chemical) and homogenized. The homogenates were centrifuged, and the protein concentrations of the collected supernatants were determined with a Pierce 660 nm Protein Assay Kit (Thermo Fisher Scientific). Then, thermal denaturation was conducted with SDS sample buffer (Thermo Fisher Scientific or Bio-Rad) containing reducing agent (Thermo Fisher Scientific). The denatured samples were loaded onto a 3–8% tris-acetate gel (Thermo Fisher Scientific) for dystrophin detection or a TGX Any kD gel (Bio-Rad, Hercules, CA) for Cas9 and GAPDH detection. Proteins were transferred to PVDF membrane by the Trans-Blot Turbo system (Bio-Rad). Blotting was performed with polyclonal anti-dystrophin (2,000-fold dilution, cat #ab15277, Abcam, Cambridge, UK), monoclonal anti-GAPDH (4,000-fold dilution, cat #2118, Cell Signaling Technology, Danvers, MA) or polyclonal anti-Cas9 (1,000-fold dilution, cat #632607, Takara Bio) and secondary anti-rabbit IgG HRP-linked whole antibody (GE Healthcare, Chicago, IL) using iBind Western Systems (Thermo Fisher Scientific) according to the manufacturer's instructions. Finally, the membranes were incubated in ECL Prime (GAPDH) or Select (dystrophin and Cas9) Western Blotting (GE Healthcare, Chicago, IL) and imaged on a ChemiDoc MP (Bio-Rad).

**Measurement of genome editing efficiency.** Genomic DNA was extracted from mouse skeletal muscle tissue (GC) by QIAamp Fast DNA Tissue Kit (Qiagen) according to the manufacturer's instructions. For T7EI (T7 Endonuclease I) assay for mouse Rosa26 sgRNA cleavage activity, target region was amplified by PCR (forward primer; 5′-CTCCGAGGCGGATCACAAGCAATAATAACCTGTAG-3′, reverse primer; 5′-TGCAAGCACGTTTCCGACTTGAGTTGCCTCAAGAG-3′) using PrimeSTAR GXL DNA polymerase (Takara bio) and the PCR amplicons were purified using QIAquick PCR purification kit (Qiagen) according to the manufacturer's instructions. Purified PCR amplicons were denatured and annealed in NEBuffer 2 (New England Biolabs, Ipswich, MA) and then digested with T7 endonuclease I (New England Biolabs) for 15 min at 37 °C. The resultant DNA fragments were analyzed by High Sensitivity D1000 Screen Tape using an Agilent 4200 TapeStation (Agilent Technologies).

For ddPCR (droplet digital PCR) analysis to assess the copy number loss at the hEx45 sgRNA #1 targeting region, genomic DNA was mixed with 2× ddPCR Supermix for Probes (no dUTP) (Bio-Rad) and two sets of primers/probes (Table 2) to amplify the sgRNA target and non-target sites. Then, droplets were generated using QX200 AutoDG (Bio-Rad) and subjected to PCR amplification and quantification using QX200 Droplet Digital PCR system (Bio-Rad).

**Immunohistochemistry for detecting dystrophin and hematoxylin and eosin (H&E) staining.** Fresh-frozen TA were sliced into 10 μm thick sections by a cryostat. The sections were reacted with 100-fold diluted polyclonal anti-dystrophin (cat #ab15277, Abcam, Cambridge, UK) or 1000-fold diluted polyclonal anti-Laminin 2 alpha (cat #ab11576, Abcam, Cambridge, UK) for 2 h at room temperature or overnight at 4 °C and then reacted with goat anti-Rabbit IgG (H;L)

**Table 2 Primers and probes for ddPCR.**

|  | Non target site | Target site |
|---|---|---|
| Forward primer | 5′-GACATGCCCATATCCAAAGGA-3′ | 5′-TTTGCCGCTGCCCAAT-3′ |
| Reverse primer | 5′-AACCGAGAGGGTGCTTTTTTC-3′ | 5′-CATTTTTGTTTTGCCTTTTTGGT-3′ |
| Probe | 5′-ACAAGACAGAAAGACACCTT-3′ | 5′-CCATCCTGGAGTTCC-3′ |
|  | VIC labeled | FAM labeled |

antibody conjugated with Alexa568 (cat #A11036, ThermoFisher, 200-fold dilution) for dystrophin detection or biotinylated IgG antibody (cat #BA-4000, Vector laboratories, 200-fold dilution) for laminin 2 alpha detection as the secondary antibody and ABC complex (Vector laboratories, CA) for 1–2 h at room temperature. The numbers of dystrophin fibers and central nuclei were counted by using HALO (Indica labs, Albuquerque, NM).

**Sequence of dystrophin gene in hEx45KI-mdx44 by Sanger method.** Reverse transcription products of GC in hEx45KI-mdx44 mice were PCR amplified with Q5 high-fidelity DNA polymerase 2× Master Mix (New England Biolabs) and forward (design of mouse *Dmd* exon 43) and reverse (design of mouse *Dmd* exon 46) primers and then purified with a QIAquick PCR Purification Kit according to the manufacturer's instructions. Purified PCR products were amplified and purified with a BigDye terminatorv3.1 Cycle sequencing Kit according to the manufacturer's instructions and sequenced with a 3500 Genetic analyzer by the Sanger method (Thermo Fisher Scientific).

**Measurement of plasma creatine kinase (CK) as DMD biomarker.** Blood was collected from the inferior vena cava and centrifuged at $20,400 \times g$ for 5 min at 4 °C. CK in collected plasma was assayed by the enzymatic method.

**Off-target analysis in vitro by the CIRCLE-seq method.** For the CIRCLE-seq analysis[36], genomic DNA was extracted from the human iPS cell line 1383D2[37]. A Covaris E210 was used to shear the purified genomic DNA to an average length of 300 bp. The seared genomic DNA was circularized and digested with Cas9 protein (Axcelead) and in vitro transcribed sgRNA in digestion buffer containing 20 mM HEPES, 150 mM KCl, 1 mM DTT, 1 mM $MgCl_2$, 50 µg/µl bovine serum albumin, and 10% glycerol. The digested DNA was processed to prepare the Illumina sequencing library by NEBNext Adaptor (cat #E7601A, NEB) ligation. The resultant libraries were quantified using a KAPA Library Quant Kit ABI Prism qPCR Mix for Illumina (KAPA biosystems) and sequenced with 75 bp paired-end reads on an Illumina NextSeq instrument.

Illumina sequencing adapter and low-quality reads (Quality score < 20) were removed from the resultant FASTQ files by cutadapt software (Version 2.4 with Python 3.4.10). Demultiplexed sequence reads were mapped on human genome hg38 with BWA-MEM (version: bwa-0.7.17-r1188). Among the mapped reads, CRISCLE-seq-specific paired-reads with opposite orientations were extracted using the 83, 163 or 99, 147 flags on SAM files, and the size of the deleted region was less than 20 bp. Filtered SAM files were converted to sorted BAM files by samtools (Version 0.1.19-44428 cd). By using bedtools (Version 2.27.1), the BAM files were converted to bedgraph files with peak height and location in the genome. Then, the peaks found in the negative control (Cas9/sgRNA untreated) were subtracted from the peaks obtained in the samples treated with each sgRNA by using bedtools with the intersect option. Finally, the common peaks between duplicated experiments were merged using bedtools intersect. The peak heights (log10) based on the chromosomal position were plotted using R (Version 3.6.1).

**Cell-based off-target analysis.** A piggyBac vector stably expressing sgRNA targeting human *DMD* exon 45 (#1) was constructed previously[37]. The sgRNA sequence against *DMD* exon 45 (#23) was inserted into the BamHI-EcoRI site of pPV-H1-ccdB-mEF1α-RiH plasmid (Addgene ID: 100598) to generate PB-H1-sgRNA-DMD23 vector. Then, we established a HEK293T cell line harboring a Dox- and dexamethasone (Dex)-inducible Cas9 CRONUS system[37] by the transfection of pPV-TetO-SphcCas9-hGR-iC-EF1a-rtTA-iP (CRONUS-Puro, Addgene ID: 100596) and PB-H1-sgRNA (DMD #1 or DMD #23) with PBase using Lipofectamine 2000 (Thermo Fisher Scientific). Two days after the transfection, puromycin and hygromycin selection was applied for two weeks to select the cells with stably integrated Cas9-GR and sgRNA. The resultant cells were treated with 6 µM Dox to induce Cas9-GR expression and 1 µM Dex to induce the nuclear shuttling of Cas9-GR for 56 days, with passaging done every 3 days. Subsequently, genomic DNA was extracted and used for T7EI assays to assess mutations at the on-target and candidate off-target sites identified by the CIRCLE-seq analysis.

**Statistical analysis.** To compare two groups, we used two-sided Aspin–Welch's t-test. To compare three groups, we used the two-sided Steel-Dwass test. $P < 0.05$ was considered statistically significant. Statistical analysis was conducted by EXSUS 2014 (Version 8.0, SAS 9.3 TS Levle1M2) from the SAS Institute, Inc. (NC).

**Reporting summary.** Further information on research design is available in the Nature Research Reporting Summary linked to this article.

## Data availability

All data supporting the findings described in this manuscript are available in the article and in the Supplementary Information, and from the corresponding author upon reasonable request. All sequence data used in the CIRCLE-seq analysis is available in the NCBI SRA database (accession number: PRJNA615771). All cell lines, plasmid vectors, and TCL053 will be available upon reasonable request under an MTA. For providing iPS cells, ethical approval of the requesting institution will be necessary. In case the donor of iPS cells opts out, we may have to stop the cell distribution, based on the informed consent. Source data are provided with this paper.

## Code availability

We only used computer software that are publicly available from the developers.

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

## Acknowledgements

We would like to thank Shinya Yamanaka, Seigo Izumo, and Yasushi Kajii for generous support and insightful discussion regarding this project, and Hidetoshi Sakurai, Keiji Itaka, Yoshitsugu Aoki, and Peter Karagiannis for their helpful advice. We also thank Yasutaka Hoashi and Yoshimasa Oomori for providing ionizable lipids, Satoshi Yamamoto and Hirokazu Matsumoto for generating hEx45KI-mdx44 mice, Hironobu Yasuno and Takeshi Watanabe for support of histological evaluation, and Megumi Ozawa, Kaori Konno, Tomomi Iwasaki, Miyuki Kobayashi, and Aya Takino for their technical assistance. This research was supported by the T-CiRA Joint Research Program (to A.H.) by Takeda Pharmaceutical Company, Ltd., and ACT-M program (to A.H. and N.I.) by AMED under Grant Number JP21im0210115.

## Author contributions

E.K. and S.M. generated the LNPs, H.H. performed the mouse experiments, Y.M. performed the in vivo Luc experiments, M.K. and Y.A. performed cytokine assay, K.A.I performed off-target detection experiments, Y.N. and A.H. performed bioinformatic analysis, M.I. and N.F. performed the DMD-iPS cell experiments, N.I. and A.H. conceived the project and supervised the team. E.K., H.H., Y.M., K.A.I., and A.H. wrote the manuscript.

## Competing interests

The authors declare the following competing interests: E.K., H.H., Y.M., S.M., M.K., Y.A., and N.I. are employees of Takeda Pharmaceutical Company, Ltd. E.K., Y.M., K.A.I., N.F., S.M., M.I., and A.H. have filed patent applications regarding the formulation of LNP and its use for delivering CRISPR. The other authors declare no competing interests.
