## [Peer Review File · Nature Communications]

Reviewers' Comments:

Reviewer #1:

Remarks to the Author:

This manuscript focuses on developing solid lipid nanoparticle formulations that can deliver mRNA to muscle tissue and induce editing in a Duchennes muscular dystrophy animal model. A new ionizable lipid is developed which can efficiently transduce muscle tissue with mRNA. Several animal experiments are presented which demonstrate that the SLN formulations can do exon skipping, after either an intramuscular injection, or limb perfusion. In addition, animal experiments were done in a humanized mouse model, which further demonstrated the ability of their SLN formulation to edit muscle tissue. Overall, this is an excellent paper and should be published in Nature Communications.

This manuscript would be strengthened by a few additional experiments. In particular, it would be helpful to do direct DNA sequencing on the muscle tissue to obtain the editing frequency, the current method in the paper appears to be based upon RNA analysis. In addition, the synthesis of the ionizable lipid should be presented, as well as its characterization. Finally, the efficacy of their ionizable lipid should be compared to commercially available solid lipid nanoparticles to gauge its improvement over existing transfection reagents.

Reviewer #2:

Remarks to the Author:

In this study, Kenjo et al described a new lipid nanoparticle (LNP) delivery system for transient delivery of CRISPR-Cas9 mRNA and sgRNA into skeletal muscle tissues. This LNP delivery system enables repeated intramuscular injections or by limb perfusion to target broad muscle tissues. Although the effect of exon 45 skipping induced by a single administration of Cas9 mRNA and sgRNA packaged in LNP is transient, repeated injections can induce stable genomic exon skipping and restore dystrophin protein in a humanized DMD mouse model. The sustained efficacy and low immunogenicity of the LNP system makes it superior to AAV delivery system and hence promising as a novel delivery vehicle to treat skeletal muscle disorders.

This study is interesting and promising in providing a new approach by combining gene editing and LNP delivery system to treat neuromuscular disorders.

The paper is well written, and methods are sound. Experiments are well designed and clearly described. Data are clearly described and analysed. I have a few comments below:

1. Page 6, line 211-214. In Fig 3 b, c and d, the hEx45KI-mdx44 mice are labelled as 'KI-mdx', while in Fig 3e,f and g they are labelled as 'hEx45KI-mdx44'. 'KI-mdx' is not mentioned or explained in Fig legend either. It would be improved by using the same label throughout Figure 3. There is NO 'Figure 3h' in Figure 3. The current Figure 3f shows GC weight, not 'calf hypertrophy', although this is described as 'figure 3g' in the main text. Please correct this part.
2. Page 7, line 230, may author provide more information on 'ASO' used to target human exon 45, i.e. what is the antisense sequence, and backbone, and the reference if it is previously reported? Since ASO approach has also been examined and compared with LNP-CRISPR in this study, may authors drop a few lines to discuss how the proposed LNP-CRISPR approach compares to the ASO exon-skipping therapy in the discussion?
3. Page 7, line 238-239. It is described in text that '..., and the dystrophin expression in TA gradually increased (Figure 4b and d)'. It is difficult to tell if Dystrophin protein is gradually increased in Fig 4d without a quantification of the western blotting data. Although from the picture dystrophin bands are denser in 6m and 12m compared to those in 0.5m, 1m and 3m, the Gapdh bands appear to show the same trend as well. A quantification of the western blotting data will help elucidate this data.
4. Page 9, line 293. It is described that 'whereas the antibody levels were much less in LNP-CRISPR-injected mice (Supplementary Fig. 8b)'. However, no Ab levels were showed (or is not detectable?) in LNP group. If this is missing, please add. If this is not detectable, please say so.
5. In the last experiment, the authors have described a widespread and efficiency of LNP-CRISPR on exon skipping at both mRNA and protein levels in different skeletal muscle groups in limbs after the intravenous limb perfusion. I wondered if authors have got a chance to measure liver function (i.e. GLDH level in plasma) in mice received 3 or 10 mg/kg LNP-CRISPR? Since liver is the major organ to metabolite LNP, this data will give a clue of the potential safety issue linked with LNP-

CRISPR skeletal muscle delivery.

Point-by-point Response Letter to the REVIEWER COMMENTS

Reviewer #1 (Remarks to the Author):

This manuscript focuses on developing solid lipid nanoparticle formulations that can deliver mRNA to muscle tissue and induce editing in a Duchennes muscular dystrophy animal model. A new ionizable lipid is developed which can efficiently transduce muscle tissue with mRNA. Several animal experiments are presented which demonstrate that the SLN formulations can do exon skipping, after either an intramuscular injection, or limb perfusion. In addition, animal experiments were done in a humanized mouse model, which further demonstrated the ability of their SLN formulation to edit muscle tissue. Overall, this is an excellent paper and should be published in Nature Communications.

We really appreciate your encouraging comments and positive feedbacks. Also, we are delighted with your endorsement of our manuscript for publication in Nature Communications.

This manuscript would be strengthened by a few additional experiments. In particular, it would be helpful to do direct DNA sequencing on the muscle tissue to obtain the editing frequency, the current method in the paper appears to be based upon RNA analysis.

To quantify the genome editing efficiency in skeletal muscle, we performed additional ddPCR analysis (Sup Fig. 6b, Sup Fig. 8a).

In addition, the synthesis of the ionizable lipid should be presented, as well as its characterization.

We added the description of the chemical synthesis procedures in the Method section.

Finally, the efficacy of their ionizable lipid should be compared to commercially available solid lipid nanoparticles to gauge its improvement over existing transfection reagents.

We have performed side-by-side comparison of our TLC053-based LNP with DLin-MC3-

DMA(MC3)-based LNP, which is one of the most widely used ionizable lipid in LNP field, by intramuscular injection into hEx45KI-mdx44 mice. We found that, in terms of delivering long Cas9 mRNA into skeletal muscle, our TCL053-LNP outperformed the efficiency of MC3-LNP, which was mainly developed for siRNA delivery into liver tissue. This data is added to the revised manuscript as new Sup Fig. 6a-c. In addition, as another benchmarking test, we obtain a commercially available in vivo lipofection reagent, called in vivo-jetRNA, which is advertised to work by intramuscular injection. As shown in Sup Fig. 6d, our TCL053-LNP showed greater genome editing activities when the same amount of Cas9 mRNA/gRNA were injected intramuscularly.

Reviewer #2 (Remarks to the Author):

In this study, Kenjo et al described a new lipid nanoparticle (LNP) delivery system for transient delivery of CRISPR-Cas9 mRNA and sgRNA into skeletal muscle tissues. This LNP delivery system enables repeated intramuscular injections or by limb perfusion to target broad muscle tissues. Although the effect of exon 45 skipping induced by a single administration of Cas9 mRNA and sgRNA packaged in LNP is transient, repeated injections can induce stable genomic exon skipping and restore dystrophin protein in a humanized DMD mouse model. The sustained efficacy and low immunogenicity of the LNP system makes it superior to AAV delivery system and hence promising as a novel delivery vehicle to treat skeletal muscle disorders.

This study is interesting and promising in providing a new approach by combining gene editing and LNP delivery system to treat neuromuscular disorders.

The paper is well written, and methods are sound. Experiments are well designed and clearly described. Data are clearly described and analysed. I have a few comments below:

We thank the reviewer for your encouraging comments. Your positive feedbacks are really appreciated by all the team members.

1. Page 6, line 211-214. In Fig 3 b, c and d, the hEx45KI-mdx44 mice are labelled as 'KI-mdx', while in Fig 3e,f and g they are labelled as 'hEx45KI-mdx44'. 'KI-mdx' is not mentioned or explained in Fig legend either. It would be improved by using the same label throughout Figure 3.

We are sorry that the label was not consistent for the hEx45KI-mdx44 mice. We corrected the labels in Fig.3 in the revised manuscript.

There is NO 'Figure 3h' in Figure 3. The current Figure 3f shows GC weight, not 'calf hypertrophy', although this is described as 'figure 3g' in the main text. Please correct this part.

Sorry for the error in figure numbering. We corrected this point in the text.

2. Page 7, line 230, may author provide more information on 'ASO' used to target human exon 45, i.e. what is the antisense sequence, and backbone, and the reference if it is previously reported? Since ASO approach has also been examined and compared with LNP-CRISPR in this study, may authors drop a few lines to discuss how the proposed LNP-CRISPR approach compares to the ASO exon-skipping therapy in the discussion?

We used phosphorodiamidate morpholino oligomer (PMO) as ASO in our experiments. We added the information about the sequence and supplier of the ASO in the Method section. In addition, we added a few lines of description to highlight the differences of ASO and LNP-CRISPR approaches (Page 11, line 364).

3. Page 7, line 238-239. It is described in text that '..., and the dystrophin expression in TA gradually increased (Figure 4b and d)'. It is difficult to tell if Dystrophin protein is gradually increased in Fig 4d without a quantification of the western blotting data. Although from the picture dystrophin bands are denser in 6m and 12m compared to those in 0.5m, 1m and 3m, the Gapdh bands appear to show the same trend as well. A quantification of the western blotting data will help elucidate this data.

As the reviewer pointed out, we realized that the band intensities of Gapdh tend to increase by the time in Fig. 4c, d. To show relative protein expression levels of Dystrophin, we added the quantification values of Dystrophin normalized by Gapdh in Fig.4 c and d. We also revised the text from "gradually increased" to "maintained" (Page 7, line 247).

4. Page 9, line 293. It is described that 'whereas the antibody levels were much less in

LNP-CRISPR-injected mice (Supplementary Fig. 8b)'. However, no Ab levels were showed (or is not detectable?) in LNP group. If this is missing, please add. If this is not detectable, please say so.

In the Sup Fig. 8b, the antibody levels in LNP-CRISPR injected group were plotted but the values were almost the same as non-injected control group and not visible. We modify the Y-axis of the graph as logarithmic scale, to clearly show the detected values.

5. In the last experiment, the authors have described a widespread and efficiency of LNP-CRISPR on exon skipping at both mRNA and protein levels in different skeletal muscle groups in limbs after the intravenous limb perfusion. I wondered if authors have got a chance to measure liver function (i.e. GLDH level in plasma) in mice received 3 or 10 mg/kg LNP-CRISPR? Since liver is the major organ to metabolite LNP, this data will give a clue of the potential safety issue linked with LNP-CRISPR skeletal muscle delivery.

We have measured the GLDH level in plasma after the limb perfusion of our LNP-CRISPR with 3 and 10 mg/kg injection dose. As shown in the new Sup Fig. 10, we observed transient elevation of GLDH level for a few days, but reverted to basal level 7 days after the LP injection. Moreover, similar elevation of GLDH level was also observed with the PBS injected group via limb perfusion, suggesting that limb perfusion procedure itself (i.e. hemostasis) may have transient liver damage. This point is added to the result section of the revised manuscript.

Reviewers' Comments:

Reviewer #1:

Remarks to the Author:

The authors have adequately addressed the comments of this reviewer. Overall, this is an excellent manuscript and is a significant contribution to the field of therapeutic genome editing.

Reviewer #2:

Remarks to the Author:

The author has addressed all my comments. There is no further comment from me.